# Adjustable Trifunctional Mid-Infrared Metamaterial Absorber Based on Phase Transition Material VO_2_

**DOI:** 10.3390/nano13121829

**Published:** 2023-06-09

**Authors:** Yi Lian, Yuke Li, Yipan Lou, Zexu Liu, Chang Jiang, Zhengda Hu, Jicheng Wang

**Affiliations:** 1Jiangsu Provincial Research Center of Light Industrial Optoelectronic Engineering and Technology, School of Science, Jiangnan University, Wuxi 214122, China; 2State Key Laboratory of Millimeter Waves, Southeast University, Nanjing 210096, China

**Keywords:** adjustable absorber, mid-infrared, phase transition, vanadium dioxide

## Abstract

In this paper, we demonstrate an adjustable trifunctional absorber that can achieve the conversion of broadband, narrowband and superimposed absorption based on the phase transition material vanadium dioxide (VO_2_) in the mid-infrared domain. The absorber can achieve the switching of multiple absorption modes by modulating the temperature to regulate the conductivity of VO_2_. When the VO_2_ film is adjusted to the metallic state, the absorber serves as a bidirectional perfect absorber with switching capability of wideband and narrowband absorption. The superposed absorptance can be generated while the VO_2_ layer is converted to the insulating state. Then, we introduced the impedance matching principle to explain the inner mechanism of the absorber. Our designed metamaterial system with a phase transition material is promising for sensing, radiation thermometer and switching devices.

## 1. Introduction

With rapid development in wireless communications and electromagnetic (EM) technologies, the EM radiation contamination problem is becoming increasingly severe. Fortunately, the EM absorber can effectively alleviate this problem. Moreover, there are promising applications of EM absorbers in the stealth technique [1], selective thermals [2], monitoring and sensing [3] and optical signal control [4]. Generally, EM absorbers can be divided into two categories: resonant absorbers and broadband absorbers. Salisbury screen and Jaumann absorbers are regarded as two classical structures of resonant absorbers [5]. The Salisbury screen, one of the first perfect absorbers, is composed of a metal–dielectric–metal (MDM) structure. Additionally, the Jaumann absorber is considered as an improvement in the Salisbury screen with more metal and dielectric layers. In addition, two typical examples of broadband absorbers are the low-density absorber and geometric transition absorber, whereby both of which depend on a high inherent loss in the component material [6]. 

In recent years, the discovery of negative index materials has opened up the exploration of metamaterials. Composed of meta units on the subwavelength scale, metamaterial can generate fantastic optical phenomena that are difficult for natural materials to realize. Furthermore, metamaterial can be utilized as a good composition material for EM absorbers. In 2008, Landy et al. proposed a microwave metamaterial absorber and introduced the metamaterial perfect absorber (MPA) for the first time, and the exploration of metamaterial absorbers in other wavelengths captured the attention of numerous researchers [7]. In 2010, Xianliang Liu et al. demonstrated an intermediate infrared (MIR) wave absorber with spatial dependence [8]. In 2011, Koray Aydin et al. proposed a nanoscale absorber in the visible range [9]. Moreover, advances in plasma and nanomanufacturing technology have led to an improvement in visible and infrared absorbers. For instance, Wang et al. presented a spiral structure metamaterial absorber in 2014, where the strong absorption was caused by multi-order plasma resonance [10]. Additionally, among the plasmonic and metamaterial absorbers, metal–insulator–metal (MIM) and MDM plasmonic metamaterial absorbers have been widely studied because of their good performance (high absorbance, polarization independence and angular insensitivity, as well as their simple manufacturing).

However, most of the proposed absorbers are non-tunable, in other words, once the structure is fixed, the absorption effect and working wavelength of absorbers are unchangeable. Recently, phase transition materials such as graphene, Ge_2_Sb_2_Te_2_ and vanadium dioxide (VO_2_) have been introduced to create adjustable optical devices. VO_2_ possesses a reversible insulating-to-metallic phase transition property that can be excited by electrical, thermal or optical simulation and has captured much attention because of its rapid response and high modulation depth. According to the photo-induced or thermal-induced phase transition characteristics of VO_2_, several tunable optical devices have been designed [11,12,13]. Tian and Li proposed an optically triggered switchable absorber based on VO_2_, which can achieve large absorptivity contrast and an ultra-fast switching rate. Additionally, an adjustable absorber based on hexagonal boron nitride and VO_2_ was designed by Song et al, whereby the absorption results can be switched from narrowband to broadband absorption via temperature control. Moreover, relying on the thermally induced phase transition of VO_2_, Zhang et al. proposed a tunable multiple broadband terahertz absorber that can achieve five broadband absorptions at 0–9.5 THz. Additionally, in this work, we utilized the thermally regulated phase change property of VO_2_ to achieve the dynamic modulation of the absorption effect. With the temperature varying from 298 K (room temperature) to 341 K, the crystal structure of VO_2_ will change from a monoclinic to a tetragonal system. Additionally, the microscopic structural change contributes to macroscopic physical property variation whereby the electrical conductivity of VO_2_ is tuned from 200 S/m to 2 × 10^5^ S/m. In general, the insulating-to-metal phase transition of VO_2_ can be achieved via just heating up and cooling down the material [14,15]. 

In our paper, we proposed a switchable absorber composed of MIM, MDM structures and a VO_2_ membrane layer that can achieve switching between three types of absorption modes, including wideband, narrowband absorption and superposed absorption of the upper and lower absorption structure. In contrast to previous tunable absorbers, our emphasis is placed on the switching of absorption modes, while most previous works can only regulate the operation frequency or improve the absorptivity [15,16]. 

## 2. Design and Theory

As is shown in Figure 1, the multilayer absorber includes the upper periodic array structure, lower MDM grating films and the VO_2_ middle membrane layer. In a sequence from the top, the absorber consists of silicon dioxide (SiO_2_) nano-pillar arrays, titanium (Ti) nano-film arrays, a magnesium fluoride (MgF_2_) layer, VO_2_ film, a zinc sulfide (ZnS) layer, zinc selenide (ZnSe) and germanium (Ge) grating membrane layers, with Plumbum (Pb) located on the bottom and in between the ZnS, ZnSe and Ge films. The structural parameters were adjusted to obtain the optimal absorption effect and they were as follows: r_1_ = 950 nm, r_2_ = 949 nm, d_1_ = 0.5d_2_, d_2_ = 1899 nm, h_1_ = 770 nm, h_2_ = 28 nm, h_3_ = 860 nm, h_4_ = 200 nm, h_5_ = 850 nm, h_6_ = 100 nm, h_7_ = 300 nm, h_8_ = 200 nm, h_9_ = 160 nm and h_10_ = 50 nm. We utilized COMSOL Multiphysics 6.0 software to conduct numerical simulation using the three-dimensional finite element method (FEM). In the modeling of the infinite arrays, periodic boundary conditions (PBCs) were set in the x and y directions and the perfectly matched layers (PMLs) were set in the z direction. The background refractive index was set as 1. The refractive indices of SiO_2_, MgF_2_ and Pb in the infrared band were taken from experimental data [17,18,19], and the permittivity of Ge, ZnS and ZnSe were taken from the previous data [20,21].

Additionally, we obtained the complex dielectric constants of Ti and VO_2_ using the Drude model:(1)$$\varepsilon (\omega )=\varepsilon_{\infty }(1-\frac{\omega_{p}^{2}}{\omega^{2}+i\omega \gamma_{p}})$$
where *ε*_∞_ represents relative dielectric constant at infinite frequency, *ω_p_* signifies the plasma frequency, *γ_p_* represents the damping coefficient, *ω* is the circular frequency of the incident planar electromagnetic wave and *ε*(*ω*) is the corresponding relative dielectric constant [19,22,23,24].

The optical characteristics of the phase transition material VO_2_ can be modulated via temperature control. The transmittance of i-VO_2_ (T = 298 K) can reach approximately 100%, while in the m-VO_2_ state (T = 341 K), the transmittance is almost zero. In other words, the VO_2_ film serves as a transparent media layer or a high-loss metal layer in insulating and metal modes, respectively. Additionally, the phase change is reversible. Therefore, the VO_2_ layer acts as a light switch that controls the optical channel between the upper and lower structures. In conclusion, the designed absorber can achieve the invertible switching of absorption modes via modulating the temperature near the insulating-to-metal state of VO_2_.

Figure 2 illustrates the three absorption modes of the absorber. When the VO_2_ film is in a metallic state (T = 341 K), the structure serves as a two-way absorber that can realize perfect broadband and narrowband absorption simultaneously, as plotted in Figure 2. When transverse magnetic (TM) waves propagate in the −z direction, the device achieves perfect absorption under an ultra-broadband wavelength ranging from 6.5 µm to 11.5 µm, with a high average absorptivity of 95.7%. Additionally, the maximal absorptivity of the wideband absorption reaches 98.5% at 9.7 µm. Additionally, when the same TM wave propagates in the +z direction, a narrowband absorption with a high-quality factor of 98.8% is obtained and the absorptivity contrast is up to 88.4% (10.34%~98.75%) at 6.5–7.5 µm. In general, when serving as a bidirectional absorber in m-VO_2_ modes (T = 341 K), the absorber can realize both broadband and narrowband absorption by altering the incidence direction, as shown in Figure 2a. Additionally, the absorber achieves a superposition absorption of wideband and narrowband absorption in the i-VO_2_ mode (T = 298 K). As demonstrated in Figure 2b, the absorptivity contrast reaches 36.1% (24.4–60.5%). The absorptivity decreases initially and then increases while the wavelength varies from 6.5 to 11.5 µm, and at the wavelength of 8 mm, the absorptivity reaches a valley of 24.4%, as shown in Figure 2b.

To gain a further understanding of the inner physical origin of the absorption, an inversion algorithm is introduced to calculate the effective impedance and absorption-related parameter. According to the effective medium theory, the complete absorption structure can be regarded as a homogeneous dielectric film. Taking case 1 as an example, the whole absorber can be regarded as a uniform film with material parameters μ_1_ and ε_1_, as demonstrated in Figure 3. Additionally, in case 2 and case 3, the structure can be deemed as a uniform layer of material parameters μ_1_′, ε_1_′ and μ_1_″, ε_1_″, respectively. For convenience, we suppose medium 1 and 3 to be air. ${\left|S_{11}\right|}^{2}$ and ${\left|S_{21}\right|}^{2}$ represent reflectance and transmittance, respectively, and can be calculated by the following formulas:(2)$$S_{11}=\frac{\Gamma_{1}(1-{\left(e^{-jk_{2}d}\right)}^{2})}{1-{\left(\Gamma_{1}e^{-jk_{2}d}\right)}^{2}}$$
(3)$$S_{21}=\frac{1-\Gamma_{1}^{2}}{1-{\left(\Gamma_{1}e^{-jk_{2}d}\right)}^{2}}e^{-jk_{2}d}$$
where *k*_2_ and Γ_1_ represent the wave number in medium 2 and the reflection coefficient between medium 1 and medium 2. Additionally, the reflection coefficient can be calculated by the following formula:(4)$$\Gamma =\frac{Z-1}{Z+1}$$
Thus, the effective impedance *Z* can be expressed by the *S*-parameters:(5)$$Z=\sqrt{\frac{{\left(1+S_{11}\right)}^{2}-S_{21}^{2}}{{\left(1-S_{11}\right)}^{2}-S_{21}^{2}}}$$
Since the transmittance reaches zero in the m-VO_2_ state, the formula can be simplified as follows:(6)$$Z=\frac{1+S_{11}}{1-S_{11}}$$
in cases 1 and 2. As for case 3, the effective impedance *Z* is calculated using Formula (5), in that the transmittance cannot be ignored in the insulating state. According to the impedance matching theory, when the normalized impedance is well matched with the vacuum impedance (*Z* = *Z*_0_), no reflectance occurs and perfect absorption is achieved. In other words, the closer the real part of the impedance is to one and the imaginary part is to zero, the higher the absorptance can reach. The scanned impedance spectra curves of case 1, 2 and 3 are illustrated in Figure 4, where the solid black and red lines represent the real and imaginary parts of the normalized impedance, and the dashed black and red lines denote the real and imaginary parts of the free space impedance, respectively. Comparing the normalized impedance curves with the absorption spectrum curves plotted in Figure 2, it is indicated that the two groups of curves are highly compatible. The numerical calculation and simulation results not only explain the physical mechanism of the absorption effect, but also verify the rationality of our design, theoretically [25,26,27].

## 3. Results and Discussion

We scanned diverse parameters to realize the optimal absorption results and obtain a better understanding of the absorption mechanism of the absorber. For convenience, we studied the cases of m-VO_2_ in the first place. As in the metallic state, the VO_2_ layer acts as a total reflective metal film; thus, there is no interference between the broadband and narrowband absorption of cases 1 and 2. Via numerical simulation, we obtained absorptivity spectrum curves and electric and magnetic field distribution maps under different polarization modes (TM and TE) of cases 1 and 2 at 9.7 µm and 9.5 µm, respectively. It is revealed in Figure 5a that the ultra-broadband absorption spectra curves and field direction are basically the same in the TM and TE modes, which results from the cross symmetry of the upper periodic structure. Additionally, as illustrated in the field direction maps in Figure 5a, an intense electric field exists on the sub-interfaces between the SiO_2_ anti-reflection layer, Ti metal film and MgF_2_ dielectric layer. At the metal–dielectric partition interface with opposite dielectric constants, surface plasmon polaritons (SPPs), the excited collective oscillation of electrons, are generated, and they play a significant role in the resonance. Additionally, there exists localized surface plasmon resonance (LSPR) in the nanostructures, whose dimensions are much smaller than the working wavelength, according to the generating principle of LSPR. Moreover, the magnetic field is mainly confined in the MgF_2_ film, indicating the presence of Fabry–Perot resonance, as plotted in Figure 5a. Inferring from the resonance theory and the field distribution, the constant ultra-wide absorption is attributed to the multiple electromagnetic resonance, which realizes light energy consumption via ohmic losses in the mediums.

As for case 2, different from the TM case, when the TE-polarized planar electromagnetic wave was vertically incident, an ultra-low absorption was observed, as shown in Figure 5b. As shown in the field direction maps, intense electric and magnetic field were mainly found in the surrounding of the top Pb metal layer in the TM mode, while there was no significant field existing in the grating structure in the TE-polarized case, at the wavelength of 9.5 µm. This is owing to the polarization sensitivity of the absorption structure based on one-dimensional grating. When the electromagnetic waves hit the grating surface under normal incidence, the electric field distribution of TM-polarized waves is vertical to the grating line, while the polarization direction of the TE wave is parallel to the grating under the same condition. Thus, the TM wave is transmitted while the TE wave is blocked under the same incident condition, in that the coupling mechanism between the incident waves and the grating structure is different in diverse polarization modes. In conclusion, the upper and lower absorption have different degrees of polarization sensitivity, which is promising for optical switching devices that enable polarization mode selection.

Considering that SPR is deeply associated with the construction parameters of the nanoscale structure, we investigated the impact of the structure parameters via numerical simulation to better understand the functionality of the absorber. The scanned absorption spectrum curves of the main parameters, including the thickness of the SiO_2_ nano-pillar h_1_, the Ti metal layer h_2_ and the MgF_2_ dielectric layer h_3_, are plotted in Figure 6, and other relevant parameters are the same as in case 1, as illustrated in Figure 2a. It is observed from Figure 6a that with the thickness of the SiO_2_ anti-reflection layer rising from 730 nm to 810 nm, the average absorptivity intensity increases initially, followed by a decreasing trend. Additionally, when the thickness of the SiO_2_ column was set as 770 nm, the stable absorptance with the highest average absorptivity was realized. Furthermore, the effect of changing the thickness of the Ti nano-cube to the absorptance is demonstrated in Figure 6b. The average absorptivity and the shape of the spectrum curve vary with the variation in h_2_. Being integrated to consider the intensity of absorptance and the stability of the spectrum curve, 28 nm was found to be the optimized parameter to achieve stable ultra-broadband absorption. As for the thickness of the MgF_2_ dielectric film, the sample interval was set as 30 nm to analyze the influence of this parameter. It was demonstrated that the influence of h_3_ to the absorptance was primarily reflected in 8.5–11.5 µm. Additionally, 860 nm is a suitable geometrical parameter that can achieve the most effective wideband absorption. The subjacent m-VO_2_ layer acts as a good barrier to obstruct electromagnetic waves, as shown in Figure 2a. Since its maximum skin depth is thinner than the layer thickness, there is no necessity to analyze the impact of its change. To analyze the influence of the intercolumnar interval r_1_ on the absorption effect, r_1_ was set as 950 nm, 951 nm, 952 nm, 953 nm, 954 nm and 955 nm. The simulated spectra curves indicate that, as the interval increases, an overall decline in the absorptance curve appears. Since the effect of changing the length of the side of the SiO_2_ nano-cube r_2_ is equivalent to r_1_, it was not further simulated. In summary, the broadband absorption results can be modulated via altering the key structure parameters. 

As for the narrowband absorption, we adjusted the main parameters of the grating cascade layers to optimize the absorption effect. Additionally, other parameters are consistent with case 2, as shown in Figure 2a. The simulated absorption spectrum curves of the key parameters, including the period d_2_ and duty cycle f, are plotted in Figure 7. The sample interval was set as 0.1 µm to carry out the simulation of the period, as shown in Figure 7a. It is demonstrated that when the grating period grows from 1.9 µm to 3.1 µm, the central wavelength red shifts. Moreover, the peak value also changes with the variation in the position. After careful observation and analysis of the regularity on the variation, it is speculated that the transformation in the absorptance is cyclical and bimodal absorption can be achieved under the wavelength of 7 µm and 11.35 µm, at the 2.3 µm case. Similarly to the period, the duty cycle also plays an indispensable role in determining the grating properties. To explore the influence of the duty cycle, we adopted a configuration that changed the slit width of the grating with the fixed period, and we set the sample interval as 0.1. It was revealed that the increase in the duty cycle of the grating led to a nearly linear red shift in the absorptivity. Moreover, in the scope of the simulation, the absorptance peak value appeared to slightly decline, which means that the absorption peak can be constantly red-shifted up to 950 nm while sustaining over 97% absorptivity via adjusting the duty cycle. In general, with the substrate layers being unchanged, the narrow absorptance peak can be modulated via just adjusting the period or duty cycle of the optical raster. By adjusting the structural parameters for numerical simulation, the tunability of the absorber and optimality of our design are further proven. To sum up, the bidirectional absorber can achieve a diverse effect of the broadband and narrowband absorption by adjusting the structure parameters; thus, the design can be used as a component of various tunable optical and optoelectronic devices such as sensors, bolometers, modulators and detectors.

As shown in Figure 8, we opted to replace the material of the critical layers to explore the influence of the material transition. Another three diverse kinds of absorption metal with high refractive index and extinction coefficient were taken to replace the Ti layer with other unchanged parameters. It is evident that Ti is the optimal material, while Zr can also be an alternative choice. The difference in the absorption spectrum curves of the different metals is associated with the absorption loss of material. Additionally, the absorption loss is in connection with loss factors of materials themselves, as well as the shape and size of the structure. In the designed absorber, Ti and Zr were proven to be the ideal absorbent materials. As for case 2, we selected four high loss metals to form the metal film layer of the grating cascade. The simulation result revealed that utilizing Pb, Al and Be can all realize a great narrow absorption effect, and choosing a Pb layer can make a balance between a high absorption peak value and a narrow width of absorption wavelength band. Moreover, according to Kirchhoff’s law, the emissivity of a material is equivalent to the absorptivity in equilibrium. Thus, the adjustability of the absorption modes is connected with the tunability of the emissivity. Since the proposed absorber can achieve broadband and narrowband absorption simultaneously, the structure paves the way for integrated optical devices, such as thermal absorber and emitter that can absorb or emit waves in specific bands or wavelengths.

The angular insensitivity of the absorber is of vital importance, in that the incident angle is difficult to be controlled accurately in the practical application. It is observed in Figure 9a,b that the broadband absorption has angular independence in both TM and TE modes and the angular tolerance can reach 60°. As for the narrow absorption of case 2, the angle insensitivity is up to 70°, and the minimum absorption peak value is still higher than 80%, while the incident angle varies from 0° to 70°. In a word, the bidirectional absorber in the m-VO_2_ state is angular-independent, which is due to the good matching of the impendence and inherent absorption of the high inherent loss metal, and this is beneficial for the practical application of absorbers.

Figure 2b illustrates the realization of the superimposed absorption in the i-VO_2_ state, indicating the implementation of optical channel switching via temperature modulation. The VO_2_ film serves as a light switch that controls the optical channel between the upper and lower structures. Here, we scanned two significant parameters, h_2_ and f, to explore the impact on the absorption effect, and other parameters were consistent with case 3, as plotted in Figure 2b. With the thickness of the MgF_2_ dielectric film rising from 10 nm to 50 nm, the absorptivity showed a decreasing trend among the wavelength range of 8–9 µm, as demonstrated in Figure 10a. Likewise, the duty cycle f was set as 0.2, 0.3, 0.4, 0.5, 0.6 and 0.7 for the simulation. Additionally, the increase in the duty cycle resulted in the red-shifting of the absorption curves, as illustrated in Figure 10b. In conclusion, by adjusting the structure parameters in the i-VO_2_ state, the effect of the superimposed absorption can also be modulated. Additionally, tunability in the three absorption patterns represents that the absorber has great application in selective and adjustable optical devices. The proposed structure opens up new avenues for integrated multifunctional optical devices, which have broad prospects.

In contrast to the previous work related to tunable absorbers based on phase transition materials, as shown in Table 1, the operation band of our proposed absorber is larger. Additionally, our designed absorber can achieve flexible switching of three absorption modes, including narrowband, broadband and superimposed absorption, while most of the previous work has focused on an adjustment in the operation band or a refinement in one kind of absorption. Additionally, the tunability means that our absorber has broad prospects in practical applications, such as optical switching devices, tunable sensing and detectors, etc.

Furthermore, we tried to implement our proposed absorber in the experiment. The VO_2_ and MgF_2_ thin films were processed on the ZnS substrate successively via sputtering deposition. The VO_2_ film with 200 nm thickness was grown on the substrates via computer-controlled pulsed DC reactive deposition from a highly purified vanadium target. Due to the complex phases of the VO_2_ material system, it is necessary to precisely control the growth temperature and optimize the sputtering gas ambient. Conventional photolithography and physical vapor deposition are applied in the fabrication of nanoscale periodic SiO_2_-Ti array structures. On the other side of the ZnS substrate, multilayer MDM grating is prepared via magnetron sputtering deposition and then etched using a focused ion beam. The multiple grid cascade structure can be processed simultaneously, which cannot only improve structural parallelism but also reduce the coating time [32,33,34,35,36].

## 4. Conclusions

In summary, we proposed a tunable triple-function absorber of a mid-infrared waveband based on the phase transition material VO_2_. Via an adjustment in temperature near the insulating-to-metal VO_2_ state, the switching of the absorption modes can be realized. When in m-VO_2_, the absorber acts as a wide-angle bidirectional absorber that can achieve ultra-wide and ultra-thin absorption at the same time, with high average absorptivity of 95.7% and 98.9%, respectively. Additionally, when in the i-VO_2_ state, superimposed absorption of the wideband and narrowband is achieved. Except for the switching of the absorption modes, the absorption effect can also be regulated via the modulation of structural parameters and component materials. The multifunctional absorber with flexible tunability has numerous application prospects in infrared sensors, polarization convertors, tunable thermal emitters and so on.

## Figures and Tables

**Figure 1 nanomaterials-13-01829-f001:**
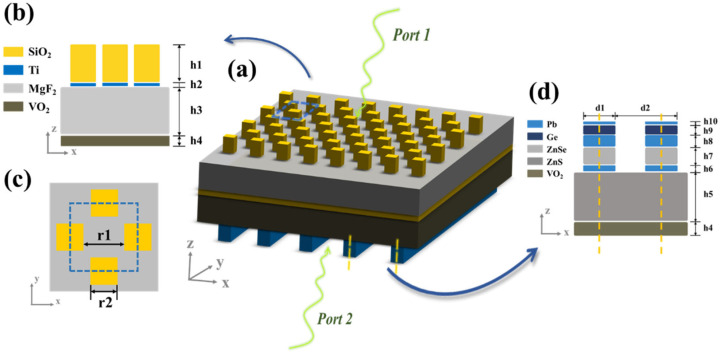
(**a**) Three-dimensional schematic illustration of the absorber. (**b**) Side view of a single cell of the upper metasurface structure. (**c**) Top view of the single cell of the upper metasurface structure. The blue dashed lines denote the region of the single cell. (**d**) Side view of a single cell of the lower grating structure. The yellow dashed lines denote the period of the grating. (r_1_ = 950 nm, r_2_ = 949 nm, d_1_ = 0.5d_2_, d_2_ = 1899 nm, h_1_ = 770 nm, h_2_ = 28 nm, h_3_ = 860 nm, h_4_ = 200 nm, h_5_ = 850 nm, h_6_ = 100 nm, h_7_ = 300 nm, h_8_ = 200 nm, h_9_ = 160 nm and h_10_ = 50 nm.).

**Figure 2 nanomaterials-13-01829-f002:**
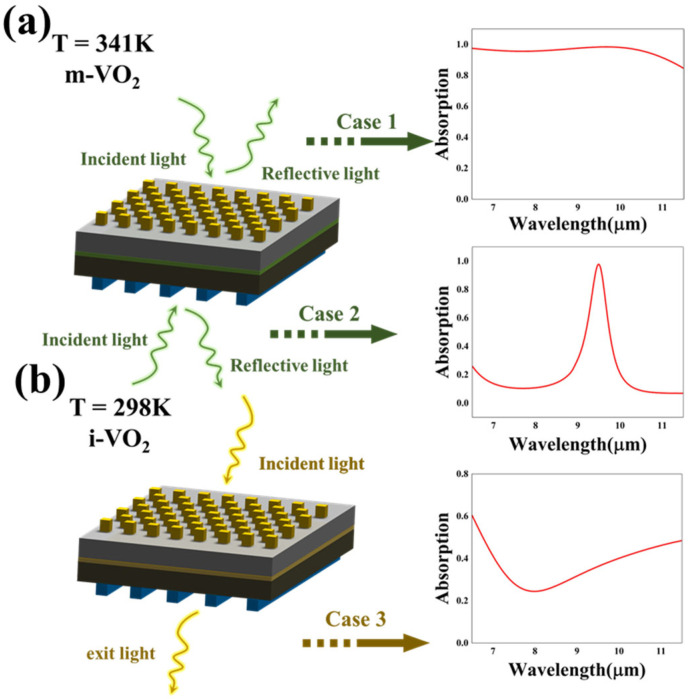
Schematic illustrations of the three absorption modes and absorption spectrum curves of case 1, case 2 (**a**) and case 3 (**b**). When a vertical TM-polarized wave strikes opposite sides of the structure in the m-VO_2_ state, the absorber realizes broadband and narrowband absorption at the same time. Additionally, when the absorber is in the i-VO_2_ state, superposed absorption is achieved.

**Figure 3 nanomaterials-13-01829-f003:**
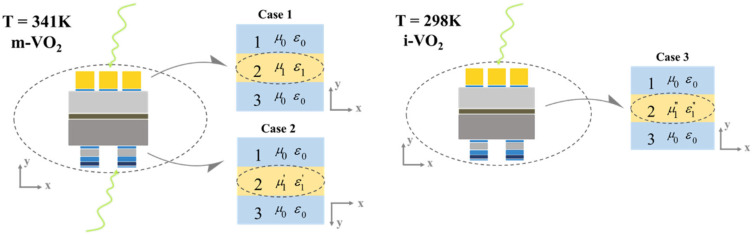
Equivalent impedance models of the absorber. According to the effective medium theory, the absorber is regarded as a uniform medium layer with material parameters μ_1_ and ε_1_. The side view of the absorber refers to Figure 1.

**Figure 4 nanomaterials-13-01829-f004:**
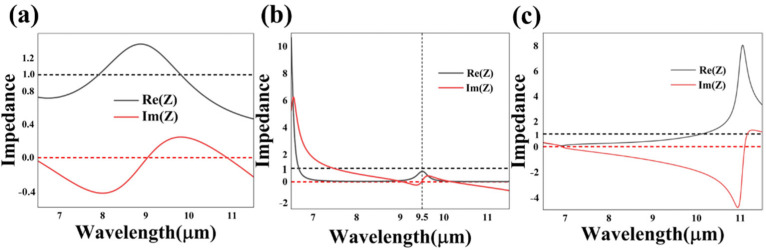
The normalized impedance curves of the proposed structure of case 1 (**a**), case 2 (**b**) and case 3 (**c**). The solid black and red curves represent the real and imaginary part of the normalized impedance, and the dashed black and red lines represent the real and imaginary part of the free space impedance, respectively.

**Figure 5 nanomaterials-13-01829-f005:**
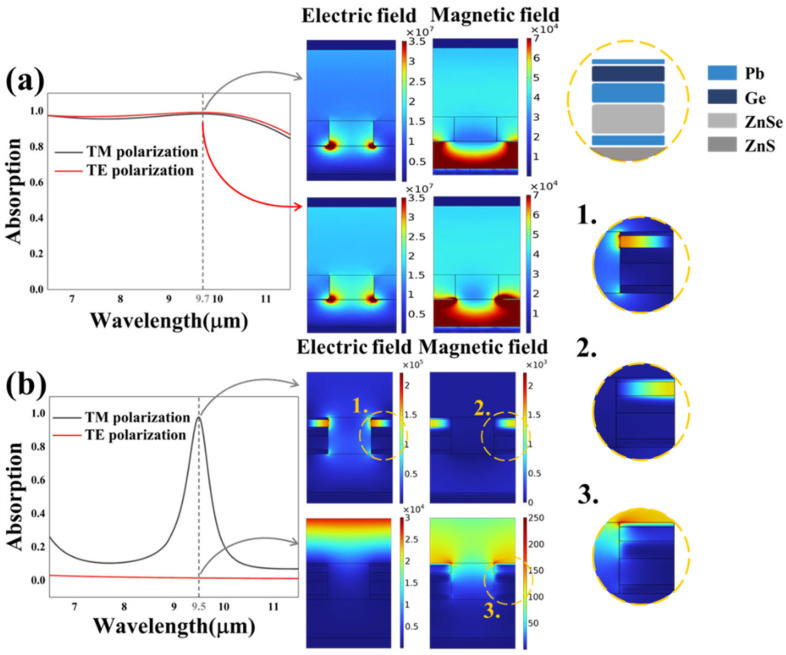
The absorption curves and electromagnetic field distribution maps of TM and TE polarization at vertical incidents of case 1 (**a**) and case 2 (**b**). The circles surrounded by the yellow dashed lines are the detailed images of field distribution.

**Figure 6 nanomaterials-13-01829-f006:**
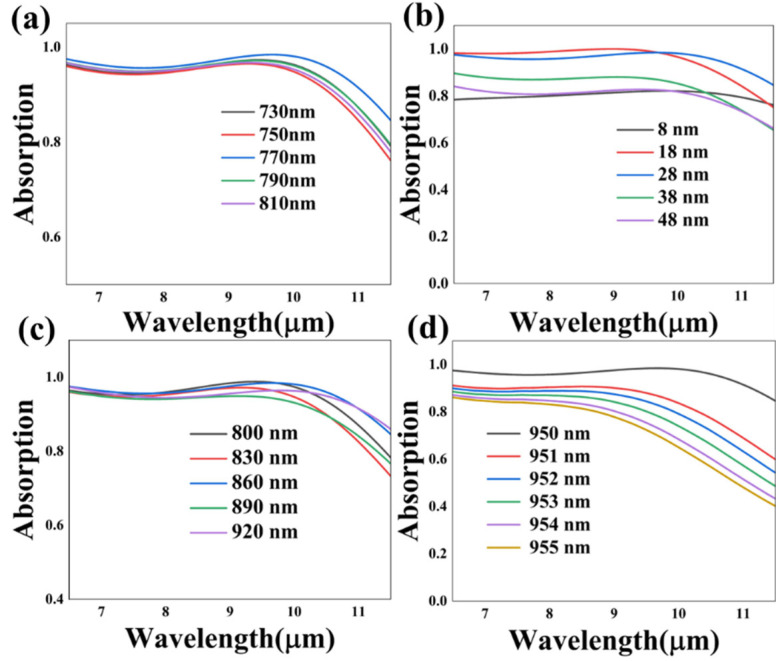
The simulated absorption spectrum curves on the different structure parameters including the thickness of the SiO_2_ nano-pillar h_1_ (**a**), the thickness of the Ti metal layer h_2_ (**b**), the thickness of the MgF_2_ dielectric layer h_3_ (**c**) and the intercolumnar interval r_1_ (**d**) of case 1.

**Figure 7 nanomaterials-13-01829-f007:**
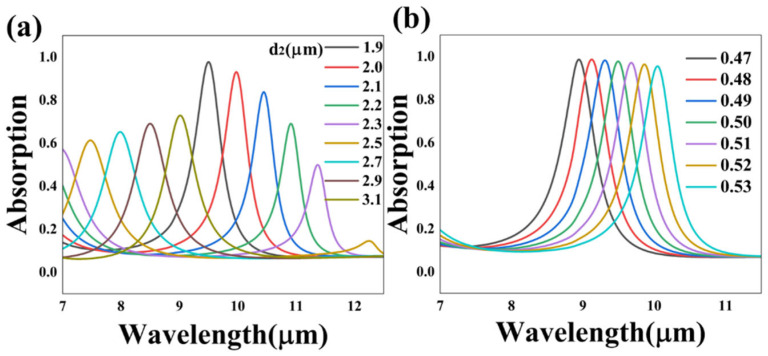
The simulated absorption spectrum curves of different grating parameters including the period d_2_ (**a**) and the duty cycle f (**b**) of case 2.

**Figure 8 nanomaterials-13-01829-f008:**
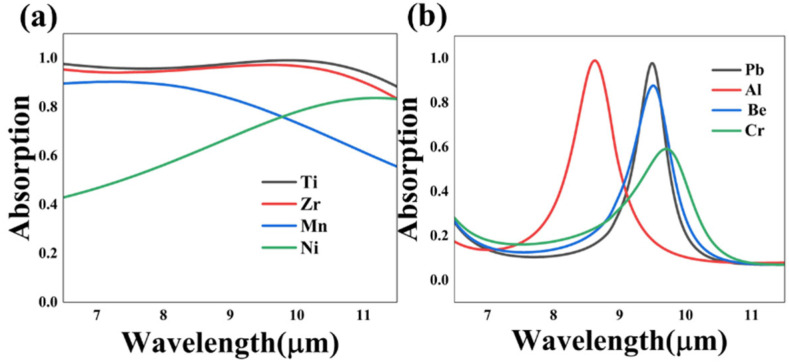
The simulated absorption spectrum curves of different materials of case 1 (**a**) and 2 (**b**).

**Figure 9 nanomaterials-13-01829-f009:**
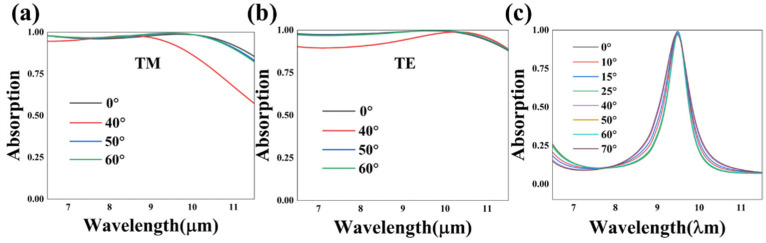
The simulated absorption spectrum curves of case 1 (**a**,**b**) and case 2 (**c**) with modulation of the incident angle. (**a**) The absorption curves of case 1 in TM modes. (**b**) The absorption curves of case 1 in TE modes. (**c**) The absorption curves of case 2 in TM modes.

**Figure 10 nanomaterials-13-01829-f010:**
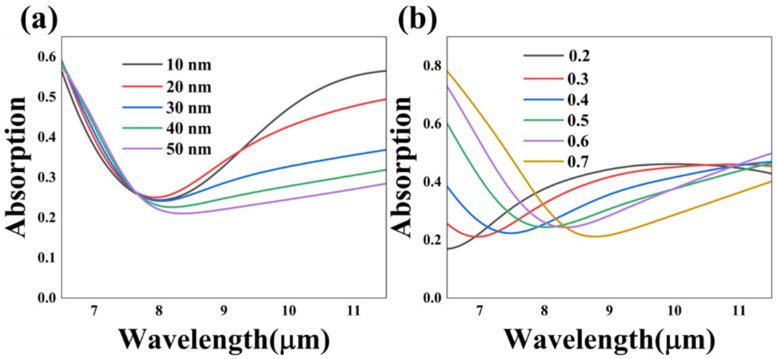
The simulated absorption spectra curves of different structure parameters including the thickness of the MgF_2_ layer h_3_ (**a**) and the grating duty cycle f (**b**) of case 3.

**Table 1 nanomaterials-13-01829-t001:** Some recent phase-transition-material-based absorbers.

Reference	Operation Band/Frequency	Absorption Property	Absorptivity	Regulation Characteristic
[28]	0.32–0.56 THz/0.356–0.682 THz	Broadband absorption	90%	Operation band tunability
[29]	10.32 GHz/9.6 GHz	Narrowband absorption	92%/91%	Operation frequency tunability
[30]	2.82 THz/2.06 THz and 3.21 THz	Single-band/dual-band absorption	98%/99.9%	Operation frequency tunability
[31]	1.566 µm/1.5–1.7 µm	Narrowband absorption/low absorption	99%/15%	Absorption mode tunability
Our work	6.5–11.5 µm	Narrowband/broadband/superimposed absorption	98.8%/95.7%/60%	Absorption mode tunability

## Data Availability

All data that support the findings of this study are included within the article.

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
