# Peer review of "Adjustable Trifunctional Mid-Infrared Metamaterial Absorber Based on Phase Transition Material VO2"

_nanomaterials, 2023, doi:10.3390/nano13121829_

Round 1

Reviewer 1 Report (Previous Reviewer 1)

Thank you for your answers. I think this manuscript is worth to publication for Nanomaterials journal.

Author Response

We thank you for your reply as well as time and effort in reviewing our manuscript. And we have carefully revised our manuscript again.

Reviewer 2 Report (Previous Reviewer 2)

The paper by Lian et al. on adjustable trifunctional mid-infrared meta material absorber based on phase-transition material VO2 does lack novelty in this form. I have serious concern regarding the approach and novelty of the manuscript. Some of my comments are listed below.

1. The author claim this proposed structure to be trifunctional, however the metamaterial design seems to be unrealistic. How one would realize the actual device experimentally? If one split the structure, i.e. the top and bottom half of the device its a simple concept of 1D grating on metallic VO2 and other would be SiO2, TiO2, MgF2, VO2 metamaterial absorber. The later hardly change with varying parameter. And the former with 1D grating is quite obvious to tune the resonance with grating pitch. So, considering the design, I don't think the proposed device is anything special or even realistic to investigate.

2. The author should ideally gshow absorbance of individual layer thin film to elaborate vividly the contribution from each layer in overall absorbance.

3. Some figure missing annotation and should be modified. 

4. Overall, I don't think the work is interesting at all and realization of the device is not realistic.

There is no novelty in the paper. I don't think it can be further considered for resubmission or major revision. The metamaterial device by the author is not realistic.

Author Response

Reviewer 3 Report (New Reviewer)

The study is completely based on numerical modeling.

Some level of experimentation would have been desirable.

A Drude model for VO2 is too simple to capture the full range of properties as a function of temperature.

English usage is quite good.

Author Response

Reviewer 4 Report (New Reviewer)

This manuscript (nanomaterials-2385783; Adjustable trifunctional mid-infrared metamaterial absorber based on phase-transition material VO2) describes a theoretical approach to utilize VO2 as a tunable triple-function absorber of mid-infrared. I recommend considering below things before publication.

Comments.

1.    In Figure 3, all layer is indicated as μ0. ε0. At least, this should be corrected. Moreover, it is not clear that which system is uniform medium layer (I guess; nano-pillar array, multilayer thin film, grating films; but not clear), please clearly identify the structure.

2.    In Figure 5a, it seems like E and H field absorption mapping based on 1 polarization direction. While, in Figure 5b, E and H field absorption mapping based on 2 polarization direction (TM, TE). It is better to show also in case of Figure 5a. Moreover, it is hard to recognize the details of multilayer structure in the Figure 5. Layer-by-layer designation is needed to enhance visibility for the readers.

3.    In Figure 8, Ti, Zr, Mn, and Ni could be categorized as Ti, Zr and Mn, Ni in terms of absorption profile. Why this difference happens? Does it affected by their loss factors?

4.    In Figure 9a and 9b, why only there is a significant difference in 40 degree incident angle?

5.    Overall, the explanation of superimposed absorption function is not sufficient. Is ~60% is meaningful?

It is a well-organized paper. English quality is fine.

Round 2

Reviewer 2 Report (Previous Reviewer 2)

Lian et.al. in their paper of "Adjustable trifunctional mid-infrared metamaterial absorber based on phase-transition material VO2" talk about use of MDM grating and metamaterial structure for broadband tunable super-absorber. The work after few revision looks interesting and complete. The author gracefully addressed all of my previous comments and hence I don't have any further comments.

Reviewer 3 Report (New Reviewer)

revised paper is improved

n/a

This manuscript is a resubmission of an earlier submission. The following is a list of the peer review reports and author responses from that submission.

Round 1

Reviewer 1 Report

The authors proposed a multi-functional metamaterials according to the VO2 phase change states. Authors simulated the bi-states VO2 ground layer to generate the absorption and transmission modes. The research topic is interesting, but needs some improvement to be published in the journal.

1) In metallic state of VO2, we can ignore the S21 in equation (5). However, we need to consider the transmittance in insulating state for case 3.

2) Proposed concept only provide the simulation results. There should be an implementation or, at least description of how to fabricated it.

3) Operation band is not clear in comparision table 1 such as reference 30, 0-5THz or unit difference.

4) I believe that the sentence "wavelength of 8 mm" is unit typo in line 121.

Reviewer 2 Report

Lian et al. has investigated numerically a platform for mid-infrared metamaterial absorber based on phase-transition material VO2. The work is solely based on simulation and a regular investigation which lack enough  novelty. In addition, I have some technical comments about the work and are listed below.

1.     Well, VO2 is a very well investigated material due to it’s phase tunability. There has been so many works on VO2 based absorber, superabsorber and broad-band absorber in mid-IR. Hence, I don’t find this work particularly carry any novelty.  Some work that the author must look through are…Plasmonic 13, 1393–1402 (2018), Xianglian Song et al 2019 J. Phys. D: Appl. Phys. 52 164002,  Optics Communications 501 (2021): 127358 etc

2.     The dual nature of absorption calculation/simulation is not conclusive for figure2a. Since i-VO2 is dielectric and how the author considers forward scattering as absorption? This is technically wrong and should be re-investigated.

3.     Moreover, there some grammatical mistake the author should emphasize.